# FREQUENCY ANALYSIS FOR GRAPH CONVOLUTION NETWORK

## ABSTRACT

In this work, we develop quantitative results to the learnability of a two-layers Graph Convolutional Network (GCN). Instead of analyzing GCN under some classes of functions, our approach provides a quantitative gap between a two-layers GCN and a two-layers MLP model. Our analysis is based on the graph signal processing (GSP) approach, which can provide much more useful insights than the message-passing computational model. Interestingly, based on our analysis, we have been able to empirically demonstrate a few cases when GCN and other state-of-the-art models cannot learn even when true vertex features are extremely low-dimensional. To demonstrate our theoretical findings and propose a solution to the aforementioned adversarial cases, we build a proof of concept graph neural network model with stacked filters named Graph Filters Neural Network (gfNN).

## 1 INTRODUCTION

Graph neural networks (GNN) is a class of neural networks which can learn from graph-structured data. Recently, graph neural networks for vertex classification and graph isomorphism test have achieved excellent results on several benchmark datasets and continuously set new state-of-the-art performance (Kipf and Welling, 2017; Veličković et al., 2019; Wu et al., 2019; Klicpera et al., 2019). Many variants of GNN have been proposed to solve problems in social networks (Hamilton et al., 2017; Zhang et al., 2018a), biology (Veličković et al., 2017; 2019), chemistry (Fout et al., 2017; Gilmer et al., 2017), natural language processing (Bastings et al., 2017; Zhang et al., 2018b; Wu et al., 2019), reasoning for vision (Santoro et al., 2017), and few/zero-shot learning (Garcia and Bruna, 2017; Kampffmeyer et al., 2019). Due to the rise of GNNs in machine learning applications, understanding GNNs theoretically has gathered lots of attention in the machine learning community.

While most theoretical results of GNNs are based on the message-passing model (Xu et al., 2019; Keriven and Peyré, 2019), there are a limited number of theoretical results for the filtering approach. However, in practice, the graph filtering view have inspired many computationally fast and high accuracy models such as ChebNet (Defferrard et al., 2016), GCN (Kipf and Welling, 2017), SplineConv (Fey et al., 2018), LanczosNet (Liao et al., 2019), and label efficient models (Li et al., 2019). In this work, we theoretically study the GCN model (Kipf and Welling, 2017) and the SGC model (Wu et al., 2019). Instead of showing which class of function a GCN or GNN model can theoretically learn, we develop quantitative bounds for the gaps between GCN/SGC and a two-layers fully connected neural network. Our theoretical results imply a few cases where SGC and GCN would fail to work: high-frequency labels, noisy features, and complex features. Interestingly, we also find other state-of-the-art GNN models such as GraphSAGE Hamilton et al. (2017) or DGI Veličković et al. (2019) failed to perform in our high-frequency labels experiments.

Recently, Wu et al. (2019) showed that graph convolutional networks (GCN) can be broken down into two steps: low-pass filtering and feature learning. Such simplification not only improves GCN's computational speed and accuracy in a wide range of problems but also allows a better understanding of the GCN model. To show the low-pass filtering nature of GCN-like models, Wu et al. (2019) used the Rayleigh quotient to bound the maximum eigenvalue when self-loops are added to the graph, which means adding self-loops created a stronger low-pass filter (Wu et al., 2019, Lemma 3, Theorem 1). Similar low-pass observation and results have also been mentioned by other works by Li et al. (2018), Qiu et al. (2018), and Li et al. (2019). We adopt SGC's two steps simplification and

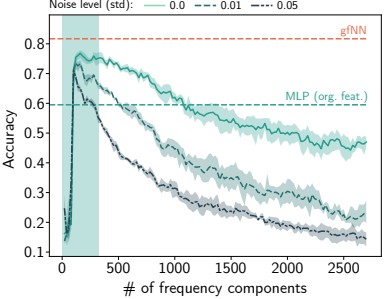

Figure 1: Accuracy by frequency components

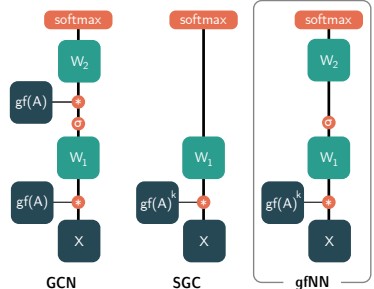

Figure 2: A simple realization of gfNN

the graph signal processing perspective (Ortega et al., 2018) to our work and further extend their implications to demonstrate when GCN or SGC does not perform well.

We claim the following contributions:

- Our first contribution is another proof for (Wu et al., 2019, Theorem 1) using the Courant-Fisher-Weyl's min-max principle. Our result (Theorem 3, Section 4) has the advantage of showing the monotone shrinking of *all* eigenvalues. In addtion, our proof on generalized eigenvalues is simpler and shorter.

- Our second and main contribution is the exploration of cases when SGC or GCN fails to work and the following performance gap Theorem:

    **Theorem 1** (Informal, see Theorem 7, 8). *Under Assumption 2, the outcomes of SGC, GCN, and gfNN are similar to those of the corresponding NNs using "true features".*

- Our third contribution is to empirically verify Assumption 2 since our Theorem 7 and 8 are based on this assumption (stated below).

**Assumption 2.** *Input features consist of frequency-limited true features and noise. The true features have sufficient information for the machine learning task.*

As a base model for analysis, we build a proof of concept model named *gfNN* (Figure 2). This model adopts the two steps simplification from SGC with two major differences: 1. We use a two-layers MLP as the classifier; and 2. We use two different filters (low-pass and high-pass). Our theorem 7 implies that, under Assumption 2, both gfNN (with appropriate filter) and GCN Kipf and Welling (2017) have similar high performance. Since gfNN does not require multiplications of the adjacency matrix during the learning phase, it is much faster than GCN. Besides, gfNN maintains an expected behavior in the aforementioned cases when other GNN models do not work.

## 2 GRAPH SIGNAL PROCESSING

Graph signal processing (GSP) regards data on the vertices as signals and applies signal processing techniques to understand the signal characteristics. By combining signals (feature vectors) and graph structure (adjacency matrix or its transformations), GSP has inspired the development of learning algorithms on graph-structured data Shuman et al. (2012). In a standard signal processing problem, it is common to assume the observations contain some noise and the underlying "true signal" has low-frequency Rabiner and Gold (1975). Our Assumption 2 is of similar nature.

Many recent GNNs were built upon results from graph signal processing. The most common practice is to multiply the *(augmented) normalized adjacency* matrix $I - \tilde{\mathcal{L}}$ with the feature matrix $\mathcal{X}$. The product $(I - \tilde{\mathcal{L}})\mathcal{X}$ is understood as features averaging and propagation. In graph signal processing literature, such operation filters signals on the graph without explicitly performing eigendecomposition on the normalized Laplacian matrix, which requires $O(n^3)$ time Vaseghi (2008). Here, we refer to this augmented normalized adjacency matrix and its variants as graph filters and propagation matrices interchangeably.

In this section, we introduce the basic concepts of graph signal processing. We adopt a recent formulation Girault et al. (2018) of graph Fourier transform on irregular graphs.

Let $\mathcal{G} = (\mathcal{V}, \mathcal{E})$ be a simple undirected graph, where $\mathcal{V} = \{1, \ldots, n\}$ be the set of $n \in \mathbb{Z}$ vertices and $\mathcal{E}$ be the set of edges.[1] Let $A = (a_{ij}) \in \mathbb{R}^{n \times n}$ be the adjacency matrix of $G$, $D = \operatorname{diag}(d(1), \ldots, d(n)) \in \mathbb{R}^{n \times n}$ be the degree matrix of $G$, where $d(i) = \sum_{j \in \mathcal{V}} a(i, j)$ is the degree of vertex $i \in \mathcal{V}$. $L = D - A \in \mathbb{R}^{n \times n}$ be the combinatorial Laplacian of $G$, $\mathcal{L} = I - D^{-1/2} A D^{-1/2}$ be the normalized Laplacian of $G$, where $I \in \mathbb{R}^{n \times n}$ is the identity matrix, and $L_{\mathrm{rw}} = I - D^{-1} A$ be the random walk Laplacian of $G$. Also, for $\gamma \in \mathbb{R}$ with $\gamma > 0$, let $\tilde{A} = A + \gamma I$ be the *augmented adjacency matrix*, which is obtained by adding $\gamma$ self loops to $G$, $\tilde{D} = D + \gamma I$ be the corresponding augmented degree matrix, and $\tilde{L} = \tilde{D} - \tilde{A} = L$, $\tilde{\mathcal{L}} = I - \tilde{D}^{-1/2} \tilde{A} \tilde{D}^{-1/2}$, $\tilde{L}_{\mathrm{rw}} = I - \tilde{D}^{-1} \tilde{A}$ be the corresponding augmented combinatorial, normalized, and random walk Laplacian matrices.

A vector $x \in \mathbb{R}^n$ defined on the vertices of the graph is called a *graph signal*. To introduce a graph Fourier transform, we need to define two operations, *variation* and *inner product*, on the space of graph signals. Here, we define the variation $\Delta \colon \mathbb{R}^n \to \mathbb{R}$ and the $\tilde{D}$-inner product by

$$\Delta(x) := \sum_{(i,j) \in \mathcal{E}} (x(i) - x(j))^2 = x^\top L x; \quad (x, y)_{\tilde{D}} := \sum_{i \in \mathcal{V}} (d(i) + \gamma) x(i) y(i) = x^\top \tilde{D} y. \quad (1)$$

We denote by $\|x\|_{\tilde{D}} := \sqrt{(x, x)_{\tilde{D}}}$ the norm induced by $\tilde{D}$. Intuitively, the variation $\Delta$ and the inner product $(\cdot, \cdot)_{\tilde{D}}$ specify how to measure the smoothness and importance of the signal, respectively. In particular, our inner product puts more importance on high-degree vertices, where larger $\gamma$ closes the importance more uniformly. We then consider the *generalized eigenvalue problem (variational form)*: Find $u_1, \ldots, u_n \in \mathbb{R}^n$ such that for each $i \in \{1, \ldots, n\}$, $u_i$ is a solution to the following optimization problem:

$$\text{minimize } \Delta(u) \text{ subject to } (u, u)_{\tilde{D}} = 1, \ (u, u_j)_{\tilde{D}} = 0, \ j \in \{1, \ldots, n\}. \quad (2)$$

The solution $u_i$ is called an $i$-th generalized eigenvector and the corresponding objective value $\lambda_i := \Delta(u_i)$ is called the $i$-th *generalized eigenvalue*. The generalized eigenvalues and eigenvectors are also the solutions to the following *generalized eigenvalue problem (equation form)*:

$$L u = \lambda \tilde{D} u. \quad (3)$$

Thus, if $(\lambda, u)$ is a generalized eigenpair then $(\lambda, \tilde{D}^{1/2} u)$ is an eigenpair of $\tilde{\mathcal{L}}$. A generalized eigenvector with a smaller generalized eigenvalue is smoother in terms of the variation $\Delta$. Hence, the generalized eigenvalues are referred to as the *frequency of the graph.*

The graph Fourier transform is a basis expansion by the generalized eigenvectors. Let $U = [u_1, \ldots, u_n]$ be the matrix whose columns are the generalized eigenvectors. Then, the *graph Fourier transform* $\mathcal{F} \colon \mathbb{R}^n \to \mathbb{R}^n$ is defined by $\mathcal{F} x = \hat{x} := U^\top \tilde{D} x$, and the *inverse graph Fourier transform* $\mathcal{F}^{-1}$ is defined by $\mathcal{F}^{-1} \hat{x} = U \hat{x}$. Note that these are actually the inverse transforms since $\mathcal{F} \mathcal{F}^{-1} = U^\top \tilde{D} U = I$.

The *Parseval identity* relates the norms of the data and its Fourier transform:

$$\|x\|_{\tilde{D}} = \|\hat{x}\|_2. \quad (4)$$

Let $h \colon \mathbb{R} \to \mathbb{R}$ be an analytic function. The *graph filter specified by $h$* is a mapping $x \mapsto y$ defined by the relation in the frequency domain: $\hat{y}(\lambda) = h(\lambda) \hat{x}(\lambda)$. In the spatial domain, the above equation is equivalent to $y = h(\tilde{L}_{\mathrm{rw}}) x$. where $h(\tilde{L}_{\mathrm{rw}})$ is defined via the Taylor expansion of $h$; see Higham (2008) for the detail of matrix functions.

In a machine learning problem on a graph, each vertex $i \in \mathcal{V}$ has a $d$-dimensional feature $\mathbf{x}(i) \in \mathbb{R}^d$. We regard the features as $d$ graph signals and define the graph Fourier transform of the features by the graph Fourier transform of each signal. Let $X = [\mathbf{x}(1); \ldots; \mathbf{x}(n)]^\top$ be the feature matrix. Then, the graph Fourier transform is represented by $\mathcal{F} X = \hat{X} =: U^\top \tilde{D} X$ and the inverse transform is $\mathcal{F}^{-1} \hat{X} = U \hat{X}$. We denote $\hat{X} = [\hat{\mathbf{x}}(\lambda_1); \ldots; \hat{\mathbf{x}}(\lambda_n)]^\top$ as the frequency components of $X$.

---

[1]We only consider unweighted edges but it is easily adopted to positively weighted edges.

## 3 EMPIRICAL EVIDENCE OF ASSUMPTION 2

The results of this paper deeply depend on Assumption 2. Thus, we first verify this assumption in real-world datasets: Cora, Citeseer, and Pubmed Sen et al. (2008). These are citation networks, in which vertices are scientific papers and edges are citations. We consider the following experimental setting: **1**. Compute the graph Fourier basis $U$ from $\tilde{\mathcal{L}}$; **2**. Add Gaussian noise to the input features: $\mathcal{X} \leftarrow \mathcal{X} + \mathcal{N}(0, \sigma^2)$ for $\sigma = \{0, 0.01, 0.05\}$; **3**. Compute the first $k$-frequency component: $\hat{\mathcal{X}}_k = U[:k]^\top \tilde{D}^{1/2}\mathcal{X}$; **4**. Reconstruct the features: $\tilde{\mathcal{X}}_k = \tilde{D}^{-1/2}U[:k]\hat{\mathcal{X}}_k$; **5**. Train and report test accuracy of a 2-layers perceptron on the reconstructed features $\tilde{\mathcal{X}}_k$. Figure 1 plots a fine-grained frequency components experiment on Cora without early stopping.

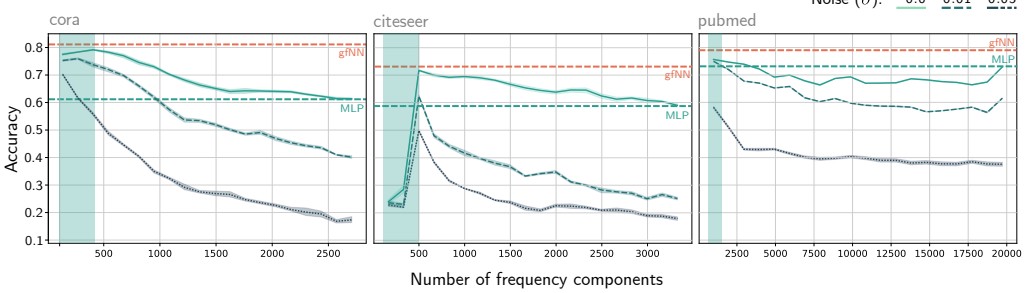

Figure 3: Average performance of 2-layers MLPs on frequency-limited feature vectors with early stopping using the validation data (`max_epochs=500`). The two horizontal lines show the performance of our model and a 2-layers MLP on original feature.

In Figure 3, we incrementally add normalized Laplacian frequency components to reconstruct feature vectors and train a 2-layers MLPs. We see that all three datasets exhibit a low-frequency nature. The classification accuracy of a 2-layers MLP tends to peak within the top 20% of the spectrum (green boxes). By adding artificial Gaussian noise, we observe that the performance at low-frequency regions is relatively robust, which implies a considerable denoising effect[2].

These experiments have confirmed our assumption that the informative features (to a given set of labels) lie in a low-frequency region in the three benchmark datasets. Note that such low-frequency nature, while coincides with low intrinsic dimensionality, has different behavior. For instance, we will show later that despite having a low intrinsic dimensionality, high-frequency sets of features and labels lead to a bad performance in both GCN and SGC. Interestingly, we observe the same performance degradation in other message-passing models like GraphSAGE (Hamilton et al., 2017).

We further confirm the low-frequency nature of the benchmark datasets' labels by computing the Rayleigh quotient for each label type in the dataset. Note that our Laplacian is normalized, the maximum value of Rayleigh quotient (corresponds to the highest eigenvalue) is 2. Figure 4 clearly shows the low-frequency nature of the labels in Cora, Citeseer, Pubmed, and Reddit datasets.

## 4 MULTIPLYING ADJACENCY MATRIX IS LOW PASS FILTERING

Computing the low-frequency components is expensive since it requires $O(|\mathcal{V}|^3)$ time to compute the eigenvalue decomposition of the Laplacian matrix. Thus, a reasonable alternative is to use a low-pass filter. Many papers on graph neural networks iteratively multiply the (augmented) adjacency matrix $\tilde{A}_{\text{rw}}$ (or $\tilde{A}$) to propagate information. In this section, we see that this operation corresponds to a *low-pass filter*.

As proven in previous work, multiplying the normalized adjacency matrix corresponds to applying graph filter $h(\lambda) = 1 - \lambda$ (Wu et al., 2019; Li et al., 2018; 2019). Since the eigenvalues of the normalized Laplacian lie on the interval $[0, 2]$, this operation resembles a *band-stop filter* that removes intermediate frequency components. However, since the maximum eigenvalue of the normalized Laplacian is 2 if and only if the graph contains a non-trivial bipartite graph as a connected component (Chung, 1997, Lemma 1.7). Therefore, for other graphs, multiplying the normalized

---

[2]The standard deviation of clean signal is less than 0.01 for all three datasets. See Table 2 for more detail.

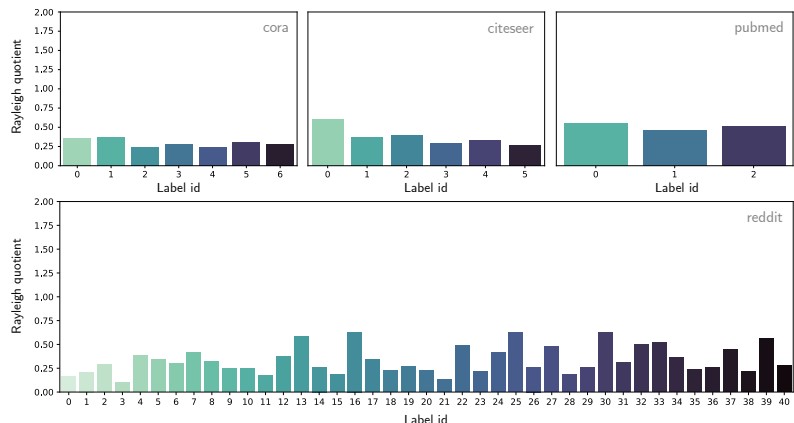

Figure 4: Rayleigh quotients of one-hot encoded label vectors for Cora, Citeseer, Pubmed, and Reddit

(non-augmented) adjacency matrix acts as a low-pass filter (i.e., high-frequency components must decrease). We can increase the low-pass filtering effect by adding self-loops (i.e., considering the augmented adjacency matrix) since it shrinks the eigenvalues toward zero as follows.[3]

**Theorem 3.** *Let $\lambda_i(\gamma)$ be the $i$-th smallest generalized eigenvalue of $(\tilde{D}, L) = (D + \gamma I)$. Then, $\lambda_i(\gamma)$ is a non-negative number, and monotonically non-increasing in $\gamma \geq 0$. Moreover, $\lambda_i(\gamma)$ is strictly monotonically decreasing if $\lambda_i(0) \neq 0$.*

Note that $\gamma$ cannot be too large. Otherwise, all the eigenvalues would be concentrated around zero, i.e., all the data would be regarded as "low-frequency"; hence, we cannot extract useful information from the low-frequency components. In practice, $\gamma$ is often set at 1 (Kipf and Welling, 2017).

From another point of view, the graph filter $h(\lambda) = 1 - \lambda$ can also be derived from a first-order approximation of a Laplacian regularized least squares Belkin and Niyogi (2004). Let us consider the problem of estimating a low-frequency true feature $\{\bar{\mathbf{x}}(i)\}_{i \in \mathcal{V}}$ from the observation $\{\mathbf{x}(i)\}_{i \in \mathcal{V}}$. Then, a natural optimization problem is given by

$$\min_{\bar{x}} \sum_{i \in \mathcal{V}} \|\bar{\mathbf{x}}(i) - \mathbf{x}(i)\|_{\tilde{D}}^2 + \Delta(\bar{X}) \tag{5}$$

where $\Delta(\bar{X}) = \sum_{p=1}^d \Delta(\bar{\mathbf{x}}(i)_p)$. Note that, since $\Delta(\bar{X}) = \sum_{\lambda \in \Lambda} \lambda \|\hat{\bar{\mathbf{x}}}(\lambda)\|_2^2$, it is a maximum a posteriori estimation with the prior distribution of $\hat{\bar{\mathbf{x}}}(\lambda) \sim N(0, I/\lambda)$. The optimal solution to this problem is given by $\bar{X} = (I + \tilde{L}_{\text{rw}})^{-1}X$. The corresponding filter is $h'(\lambda) = (1 + \lambda)^{-1}$, and hence $h(\lambda) = 1 - \lambda$ is its first-order Taylor approximation.

## 5 BIAS-VARIANCE TRADE-OFF FOR LOW PASS FILTERS

In the rest of this paper, we establish theoretical results under Assumption 2. To be concrete, we pose a more precise assumption as follows.

**Assumption 4** (Precise Version of the First Part of Assumption 2). *Observed features $\{\mathbf{x}(i)\}_{i \in \mathcal{V}}$ consists of true features $\{\bar{\mathbf{x}}(i)\}_{i \in \mathcal{V}}$ and noise $\{\mathbf{z}(i)\}_{i \in \mathcal{V}}$. The true features $\bar{X}$ have frequency at most $0 \leq \epsilon \ll 1$ and the noise follows a white Gaussian noise, i.e., each entry of the graph Fourier transform of $Z$ independently identically follows a normal distribution $N(0, \sigma^2)$.*

Using this assumption, we can evaluate the effect of the low-pass filter as follows.

**Lemma 5.** *Suppose Assumption 4. For any $0 < \delta < 1/2$, with probability at least $1 - \delta$, we have*

$$\|\bar{X} - \tilde{A}_{rw}^k X\|_D \leq \sqrt{k}\epsilon \|\bar{X}\|_D + O\left(\sqrt{\log(1/\delta)R(2k)}\right) \mathbb{E}[\|Z\|_D], \tag{6}$$

---

[3]The shrinking of the maximum eigenvalue has already been proved in (Wu et al., 2019, Theorem 1). Our theorem is stronger than theirs since we show the "monotone shrinking" of "all" the eigenvalues. In addition, our proof is simpler and shorter.

*where $R(2k)$ is a probability that a random walk with a random initial vertex returns to the initial vertex after $2k$ steps.*

The first and second terms of the right-hand side of equation 6 are the *bias term* incurred by applying the filter and the *variance term* incurred from the filtered noise, respectively. Under the assumption, the bias increases a little, say, $O(\sqrt{\epsilon})$. The variance term decreases in $O(1/\deg^{k/2})$ where $\deg$ is a typical degree of the graph since $R(2k)$ typically behaves like $O(1/\deg^k)$ for small $k$.[4] Therefore, we can obtain a more accurate estimation of the true data from the noisy observation by multiplying the adjacency matrix if the maximum frequency of $\bar{X}$ is much smaller than the *noise-to-signal ratio* $\|Z\|_D/\|\bar{X}\|_D$.

This theorem also suggest a choice of $k$. By minimizing the right-hand side by $k$, we obtain:

**Corollary 6.** *Suppose that $\mathbb{E}[\|Z\|_D] \leq \rho\|\bar{X}\|_D$ for some $\rho = O(1)$. Let $k^*$ be defined by $k^* = O(\log(\log(1/\delta)\rho/\epsilon))$, and suppose that there exist constants $C_d$ and $\bar{d} > 1$ such that $R(2k) \leq C_d/\bar{d}^k$ for $k \leq k^*$. Then, by choosing $k = k^*$, the right-hand side of equation 6 is $\tilde{O}(\sqrt{\epsilon})$.[5]* $\square$

This concludes that we can obtain an estimation of the true features with accuracy $\tilde{O}(\sqrt{\epsilon})$ by multiplying $\tilde{A}_{\mathrm{rw}}$ several times.

## 6 GRAPH FILTER NEURAL NETWORK

In the previous section, we see that low-pass filtered features $\tilde{A}_{\mathrm{rw}}^k X$ are accurate estimations of the true features with high probability. In this section, we analyze the performance of this operation. In practice, $k = 2$ is sufficient for most datasets. Therefore, for simplicity, we set $k = 2$.

Let the multi-layer (two-layer) perceptron be

$$h_{\mathrm{MLP}}(X|W_1, W_2) = \sigma_2(\sigma_1(XW_1)W_2), \tag{7}$$

where $\sigma_1$ is the entry-wise ReLU function, and $\sigma_2$ is the softmax function. Note that both $\sigma_1$ and $\sigma_2$ are contraction maps, i.e., $\|\sigma_i(X) - \sigma(Y)\|_D \leq \|X - Y\|_D$.

**gfNN** As a toy model for analysis and experiments, we define our gfNN models. It comes with two flavors: low-frequency and high-frequency. In this section, we provide the analysis for the low-frequency version and emprical results (Section 7) for high-frequency version.

$$h_{\mathrm{gfNN-low}}(X, A|W_1, W_2) = h_{\mathrm{MLP}}(\tilde{A}_{\mathrm{rw}}^2 X|W_1, W_2) \tag{8}$$

$$h_{\mathrm{gfNN-high}}(X, A|W_1, W_2) = h_{\mathrm{MLP}}(\tilde{L}_{\mathrm{rw}}^2 X|W_1, W_2) \tag{9}$$

Under the second part of Assumption 2, our goal is to obtain a result similar to $h(\bar{X}|W_1, W_2)$. The simplest approach is to train a multi-layer perceptron $h_{\mathrm{MLP}}(X|W_1, W_2)$ with the observed feature. The performance of this approach is evaluated by

$$\|h_{\mathrm{MLP}}(\bar{X}|W_1, W_2) - h_{\mathrm{MLP}}(X|W_1, W_2)\|_{\tilde{D}} \leq \|Z\|_D\rho(W_1)\rho(W_2), \tag{10}$$

where $\rho(W)$ is the maximum singular value of $W$.

Now, we consider applying graph a filter $\tilde{A}_{\mathrm{rw}}^k$ to the features, then learning with a multi-layer perceptron, i.e., $h_{\mathrm{MLP}}(h(\tilde{L}_{\mathrm{rw}})X|W_1, W_2)$. Using Lemma 5, we obtain the following result.

**Theorem 7.** *Suppose Assumption 4 and conditions in Corollary 6. By choosing $k$ as Corollary 6, with probability at least $1 - \delta$ for $\delta < 1/2$,*

$$\|h_{MLP}(\bar{X}|W_1, W_2) - h_{MLP}(\tilde{A}_{rw}^2 X|W_1, W_2)\|_D = \tilde{O}(\sqrt{\epsilon})\mathbb{E}[\|Z\|_D]\rho(W_1)\rho(W_2). \tag{11}$$

This means that if the maximum frequency of the true data is small, we obtain a solution similar to the optimal one.

---

[4]This exactly holds for a class of locally tree-like graphs Dembo et al. (2013), which includes Erdos–Renyi random graphs and the configuration models.

[5]$\tilde{O}(\cdot)$ suppresses logarithmic dependencies.

**Comparison with GCN**   Under the same assumption, we can analyze the performance of a two-layers GCN. The GCN is given by

$$h_{\text{GCN}}(X|W_1, W_2) = \sigma_2(\tilde{A}_{\text{rw}}\sigma_1(\tilde{A}_{\text{rw}}XW_1)W_2). \tag{12}$$

**Theorem 8.** *Under the same assumption to Theorem 7, we have*

$$\|h_{MLP}(\bar{X}|W_1, W_2) - h_{GCN}(X|W_1, W_2)\|_D \leq \tilde{O}(\sqrt[4]{\epsilon})\mathbb{E}[\|Z\|_D]\rho(W_1)\rho(W_2). \tag{13}$$

This theorem means that GCN also has a performance similar to MLP for true features. Hence, in the limit of $\epsilon \to 0$ and $Z \to 0$, all MLP, GCN, and the gfNN have the same performance.

In practice, gfNN has two advantages over GCN. First, gfNN is faster than GCN since it does not use the graph during the training. Second, GCN has a risk of overfitting to the noise. When the noise is so large that it cannot be reduced by the first-order low-pass filter, $\tilde{A}_{\text{rw}}$, the inner weight $W_1$ is trained using noisy features, which would overfit to the noise leading to low test accuracy.

**Comparison with SGC**   Recall that a degree-2 SGC is given by

$$h_{\text{SGC}}(X|W_1) = \sigma_2(\tilde{A}_{\text{rw}}^2 X W_2), \tag{14}$$

i.e., it is a gfNN with a one-layer perceptron. Thus, by the same discussion, SGC has a performance similar to the perceptron (instead of the MLP) applied to the true feature. This clarifies an issue of SGC: if the true features are non-linearly separable, SGC cannot learn the correct function.

## 7    EXPERIMENTS

We first show the advantage of models like gfNN-low and SGC over GCN by injecting Gaussian noise to the three benchmark citation networks. In accordance with Theorem 8, we see that GCN-like models have low noise tolerance while gfNN-low and SGC are much more robust. Our experiments on citation networks are conducted in the same *transductive* setting used by Kipf and Welling (2017) and Wu et al. (2019). Training details are provided in Appendix B and C.

Experimental results in Figure 5 confirms our hypothesis for the three benchmark datasets. We see that when a high level of noise is injected to low-frequency datasets (especially Cora), the performance of GCN can degrade much faster than SGC or gfNN. It is also trivial to see that Logistic Regression (LG) and MLP on raw features exhibit similar behavior when noise is introduced.

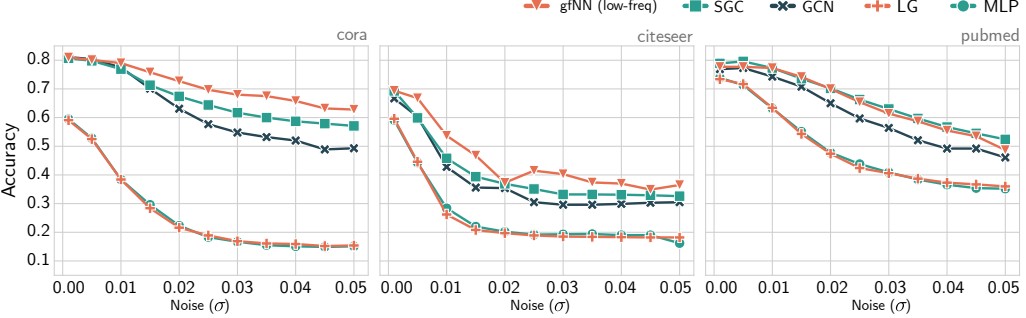

Figure 5: Benchmark test accuracy on Cora (left), Citeseer (middle), and Pubmed (right) datasets. The noise level is measured by standard deviation of white noise added to the features.

Next, we experiment on two synthetic datasets: Two Circles and BA-High. These datasets are synthesized to show: 1. SGC cannot learn non-linearly separable features; and 2. SGC, GCN, DGI (Veličković et al., 2019), and GraphSAGE (Hamilton et al., 2017) fail when it comes to high-frequency labels (even when the features are informative). Two circles dataset is generated by generating $4,000$ points sampled from two nested circles and define a 5-nearest neighbors graph on the data points. Visualization of this dataset is given in Appendix E. We generate BA-High dataset by constructing a BA-graph with $n = 200$ and $n = 10$ (200 nodes, every new node are attached to 10 neighbors

selected from existing nodes). We then find the maximal independent set for the graph and use this set as label 1. The rest of the nodes are label 0. For each node with label 1, we generate a 50-dimensions feature vector from a Gaussian $\mathcal{N}(+\epsilon, 1.0)$. We do the same for all nodes with label 0 but with Gaussian $\mathcal{N}(-\epsilon, 1.0)$. In Table 1 we report the results at $\epsilon = 0.5$.

Unsurprisingly, SGC cannot learn the non-linearly separable feature in the Two Circle experiment (Table 1) while other more complex models perform as expected (similar performance as an MLP). More interesting is the results of the BA-High experiment (Table 1, Figure 6). Surprisingly, we find all popular GNN models cannot quite learn under this simple setting. Although the node features are informative (a simple LogReg or MLP can have accuracy larger than 90%), most GNN models reported less than 60% accuracy. In Figure 6, we show a simple solution to this setting: Use high frequency eigenvectors or high-pass filter (gfNN-High).

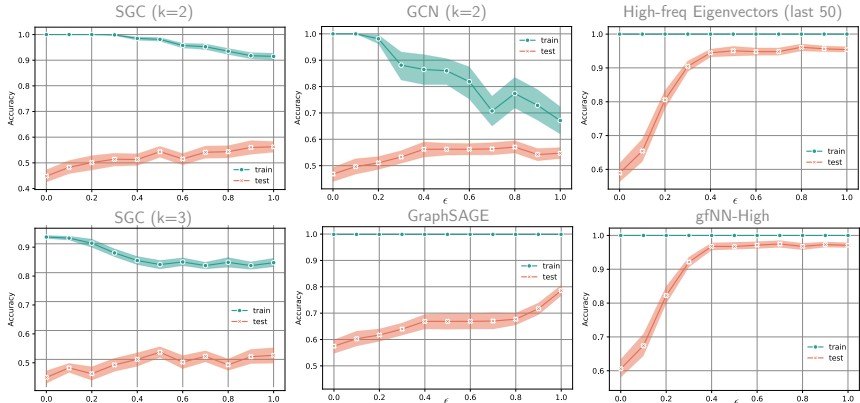

Figure 6: Performance on high-frequency labels

For comparison purpose, we present our results on benchmark datasets in Table 1. Model details are presented in Appendix C. Dataset details are presented in Table 2 (Appendix E).

Table 1: Average test accuracy on original train/val/test splits (50 times)

|  | Cora | Citeseer | Pubmed | Reddit | PPI | 2Circles | BA-High |
|---|---|---|---|---|---|---|---|
| **DGI** | $83.1 \pm 0.2$ | $72.1 \pm 0.1$ | $80.1 \pm 0.2$ | $94.5 \pm 0.3$ | $99.2 \pm 0.1$ | $85.2 \pm 0.6$ | $54.6 \pm 1.8$ |
| **GCN** | $80.0 \pm 1.8$ | $69.6 \pm 1.1$ | $79.3 \pm 1.3$ | - | - | $84.9 \pm 0.8$ | $58.9 \pm 2.2$ |
| **SGC** | $77.6 \pm 2.2$ | $65.6 \pm 0.1$ | $78.4 \pm 1.1$ | $94.9 \pm 0.2$ | $89.0 \pm 0.1$ | $53.5 \pm 1.4$ | $55.5 \pm 1.3$ |
| **gfNN-low** | $82.3 \pm 0.2$ | $71.8 \pm 0.1$ | $79.2 \pm 0.2$ | $94.8 \pm 0.2$ | $89.3 \pm 0.5$ | $85.6 \pm 0.8$ | $55.4 \pm 2.3$ |
| **gfNN-high** | $24.2 \pm 1.9$ | $22.5 \pm 2.2$ | $43.6 \pm 1.3$ | $10.5 \pm 2.6$ | $86.6 \pm 0.1$ | $48.3 \pm 3.5$ | $96.2 \pm 1.0$ |

## 8 DISCUSSION

Recently, Kampffmeyer et al. (2019) observed that GCN models might have too strong Laplacian smoothing effect (low-frequency in our terms) for zero-shot learning application. This is an example of an application where the frequency of data might be band-limited or higher than benchmark graph datasets. Some other possible scenario in a social network might include more complex and high-frequency nature that GNN models like GCN cannot learn well. One possible solution to this scenario is to use both low- and high-pass filters like gfNN and select the model by validation data or learn from both filters.

Our analysis and experimental results showed that GNN models work well on benchmark datasets because these datasets have the low-frequency nature in both feature vectors and labels. With high-frequency labels, popular GNN models will perform badly. While there is not yet any real-world benchmark dataset which node labels are of high-frequency nature, such setting is not entirely impossible given the complexity of a network. For instance, a relationship network would be less likely to have same-gender links. Therefore, gender labels might exhibit high-frequency nature.

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

APPENDIX

## A    PROOFS

*Proof of Theorem 3.* Since the generalized eigenvalues of $(D + \gamma I, L)$ are the eigenvalues of a positive semidefinite matrix $(D + \gamma I)^{1/2} L (D + \gamma I)^{1/2}$, these are non-negative real numbers. To obtain the shrinking result, we use the Courant–Fisher–Weyl's min-max principle (Bhatia, 2013, Corollary III. 1.2): For any $0 \le \gamma_1 < \gamma_2$,

$$\lambda_i(\gamma_2) = \min_{H:\text{subspace},\dim(H)=i} \max_{x \in H, x \ne 0} \frac{x^\top L x}{x^\top (D + \gamma_2 I) x} \tag{15}$$

$$\le \min_{H:\text{subspace},\dim(H)=i} \max_{x \in H, x \ne 0} \frac{x^\top L x}{x^\top (D + \gamma_1 I) x} \tag{16}$$

$$= \lambda_i(\gamma_1). \tag{17}$$

Here, the second inequality follows because $x^\top (D + \gamma_1)x < x^\top (D + \gamma_2)x$ for all $x \ne 0$ Hence, the inequality is strict if $x^\top L x \ne 0$, i.e., $\lambda_i(\gamma_1) \ne 0$. □

**Lemma 9.** *If $X$ has a frequency at most $\epsilon$ then $\|h(\mathcal{L})X\|_D^2 \le \max_{t \in [0,\epsilon]} h(t)\|X\|_D^2$.*

*Proof.* By the Parseval identity,

$$\|\mathcal{L}X\|_D^2 = \sum_{\lambda \in \Lambda} h(\lambda)\|\hat{\mathbf{x}}(\lambda)\|_2^2 \tag{18}$$

$$\le \max_{t \in [0,\epsilon]} h(t) \sum_{\lambda \in \Lambda} \|\hat{\mathbf{x}}(\lambda)\|_2^2 \tag{19}$$

$$= \max_{t \in [0,\epsilon]} h(t)\|\hat{X}\|_2^2 \tag{20}$$

$$= \max_{t \in [0,\epsilon]} h(t)\|\hat{X}\|_D^2. \tag{21}$$

□

*Proof of Lemma 5.* By substituting $X = \bar{X} + Z$, we obtain

$$\|\bar{X} - \tilde{A}_{\text{rw}}^k X\|_{\tilde{D}} \le \|(I - \tilde{A}_{\text{rw}}^k)\bar{X}\|_{\tilde{D}} + \|\tilde{A}_{\text{rw}}^k Z\|_{\tilde{D}}. \tag{22}$$

By Lemma 9, the first term is bounded by $\sqrt{k}\epsilon\|\bar{X}\|_D$. By the Parseval identity equation 4, the second term is evaluated by

$$\|\tilde{A}_{\text{rw}}^k Z\|_{\tilde{D}}^2 = \sum_{\lambda \in \Lambda} (1 - \lambda)^{2k}\|\hat{\mathbf{z}}(\lambda)\|_2^2 \tag{23}$$

$$= \sum_{\lambda \in \Lambda, p \in \{1,\dots,d\}} (1 - \lambda)^{2k}\hat{z}(\lambda, p)^2 \tag{24}$$

By (Laurent and Massart, 2000, Lemma 1), we have

$$\text{Prob}\left\{ \sum_{\lambda,p} (1 - \lambda)^{2k}(\hat{z}(\lambda, p)^2/\sigma^2 - 1) \ge 2\sqrt{t \sum_{\lambda,p} (1 - \lambda)^{4k}} + 2t \right\} \le e^{-t}. \tag{25}$$

By substituting $t = \log(1/\delta)$ with $\delta \le 1/2$, we obtain

$$\text{Prob}\left\{ \sum_{\lambda,p} (1 - \lambda)^{2k}\hat{z}(\lambda, p)^2 \ge 5\sigma^2 nd \log(1/\delta)\frac{\sum_\lambda (1 - \lambda)^{2k}}{n} \right\} \le 1 - \delta. \tag{26}$$

Note that $\mathbb{E}[\|Z\|_D^2 = \sigma^2 nd$, and

$$\frac{\sum_\lambda (1 - \lambda)^{2k}}{n} = \frac{\text{tr}(\tilde{A}_{\text{rw}}^{2k})}{n} = R(2k) \tag{27}$$

since $(i, j)$ entry of $\tilde{A}_{\text{rw}}^{2k}$ is the probability that a random walk starting from $i \in \mathcal{V}$ is on $j \in \mathcal{V}$ at $2k$ step, we obtain the lemma. □

*Proof of Theorem 7.* By the Lipschitz continuity of $\sigma$, we have

$$\|h_{\text{MLP}}(\bar{X}|W_1, W_2) - h_{\text{MLP}}(\tilde{A}_{\text{rw}}^k X|W_1, W_2)\|_D \leq \|\bar{X} - \tilde{A}_{\text{rw}}^k X\|_D \rho(W_1)\rho(W_2). \tag{28}$$

By using Lemma 5, we obtain the result. □

**Lemma 10.** *If $X$ has a frequency at most $\epsilon$ then there exists $Y$ such that $Y$ has a frequency at most $\sqrt{\epsilon}$ and $\|\sigma(X) - Y\|_D^2 \leq \sqrt{\epsilon}\|X\|_D^2$.*

*Proof.* We choose $Y$ by truncating the frequency components greater than $\sqrt{\epsilon}$. Then,

$$\|\sigma(X) - Y\|_D^2 = \sum_{\lambda > \sqrt{\epsilon}} \|\widehat{\sigma(X)}(\lambda)\|^2 \tag{29}$$

$$\leq \frac{1}{\sqrt{\epsilon}} \sum_{\lambda} \lambda\|\widehat{\sigma(X)}(\lambda)\|^2 \tag{30}$$

$$= \frac{1}{\sqrt{\epsilon}} \sum_{(u,v)\in E} \|\sigma(x(u)) - \sigma(x(v))\|_2^2 \tag{31}$$

$$\leq \frac{1}{\sqrt{\epsilon}} \sum_{(u,v)\in E} \|x(u) - x(v)\|_2^2 \tag{32}$$

$$= \sqrt{\epsilon} \sum_{\lambda} \lambda\|\hat{x}(\lambda)\|_2^2 \tag{33}$$

$$\leq \sqrt{\epsilon} \sum_{\lambda} \|\hat{x}(\lambda)\|_2^2 \tag{34}$$

$$= \sqrt{\epsilon}\|X\|_D^2. \tag{35}$$

□

*Proof of Theorem 8.* From Lemma 10, by taking the square root and absolute values of both sides, we have $\|\sigma(X) - Y\|_D \leq \epsilon^{1/4}\|X\|_D$.

By setting $Y = \sigma(\tilde{A}_{\text{rw}}\bar{X})$, since $\sigma$ is an activation function, we have:

$$\|\sigma(\bar{X}) - \tilde{A}_{\text{rw}}\sigma(\bar{X})\|_D = \|\tilde{L}_{\text{rw}}\sigma(\bar{X})\|_D \tag{36}$$

$$\leq O(\epsilon^{1/4})\|\bar{X}\|_D \tag{37}$$

Consequently, we consider the D-norm stated in Theorem 8:

$$\|\sigma(\bar{X}W_1)W_2 - \tilde{A}_{\text{rw}}\sigma(\tilde{A}_{\text{rw}}XW_1)W_2\|_D \tag{38}$$

$$\leq \left(\|\sigma(\bar{X}W_1) - \tilde{A}_{\text{rw}}\sigma(\bar{X}W_1)\|_D + \|\tilde{A}_{\text{rw}}\sigma(\bar{X}W_1) - \tilde{A}_{\text{rw}}\sigma(\tilde{A}_{\text{rw}}XW_1)\|_D\right)\rho(W_2) \tag{39}$$

$$= \left(\|\tilde{L}_{\text{rw}}\sigma(\bar{X}W_1)\|_D + \|\bar{X} - \tilde{A}_{\text{rw}}X\|_D\rho(W_1)\right)\rho(W_2) \tag{40}$$

$$\leq \left(O(\epsilon^{1/4})\|\bar{X}\|_D + R(2)\|Z\|_D\right)\rho(W_1)\rho(W_2). \tag{41}$$

□

## B GFNN MODEL DETAILS

Our gfNN model is implemented in Pytorch. While other number of hidden layers are possible, we find that 1 hidden layer with 256 units (2 layers MLP) is enough for all benchmark datasets. Other changes does not significantly influence the experimental results. It worth mentioning that each gfNN model actually has 2 identical neural networks inside, one to learn from low-pass filter $\tilde{A}_{\text{rw}}$ and the other for high-pass filter $\tilde{L}_{\text{rw}}$. We use PReLU (0.25) as the activation function since there are a slight improvement (about 1%) in validation accuracy when training with PReLU compare to ReLU.

To optimize our model, we use the Pytorch built-in Adam optimizer. We fix the hyperparameters accross all datasets:

- Weight decay: $1 \times 10^{-4}$
- Learning rate: 0.01
- Batch size: 32
- Early stopping: 40 epochs patience (Max. 500 epochs).

All datasets have three sets: train, valdiation, and test. The validation set is used for early stopping and model selection (high/low). The test accuracy is reported for all experiments.

## C  BASELINE DETAILS

We set GCN (Kipf and Welling, 2017) at two layers with 64 hidden units, ReLU activation, no dropout. We keep the setting of SGC (Wu et al., 2019) as recommended by the original implementation (200 epochs, lr=0.1, weight decay: $1 \times 10^{-4}$, ReLU activation). For DGI (Veličković et al., 2019), GraphSAGE Hamilton et al. (2017), and GIN (Xu et al., 2019), we follow the authors' default paramter tuning. The result for GraphSAGE is reported as the best result from "small" models (mean, sum, seq) for 200 epochs training, we also set number of level-1 neighbors at 25 and level-2 at 10.

## D  DATASET DETAILS

We provide an evaluation for the semi-supervised node classification performance of SGC on the Cora, Citeseer, and Pubmed citation network datasets Sen et al. (2008). We also follow Wu et al. (2019) to run experiments on Reddit Chen et al. (2018) and Hamilton et al. (2017) on PPI Zitnik and Leskovec (2017).

Table 2: Real-world benchmark datasets and synthetic datasets for vertex classification

| Dataset | Nodes | Edges | Features ($X$) | $(\mu_X, \sigma_X)$ | Classes | Train/Val/Test |
|---|---|---|---|---|---|---|
| **Cora** | 2,708 | 5,278 | 1,433 | (0.0007, 0.0071) | 7 | 140/500/1,000 |
| **Citeseer** | 3,327 | 4,732 | 3,703 | (0.0003, 0.0029) | 6 | 120/500/1,000 |
| **Pubmed** | 19,717 | 44,338 | 500 | (0.0019, 0.0087) | 3 | 60/500/1,000 |
| **Reddit** | 231,443 | 11,606,919 | 602 | - | 41 | 151,708/23,699/55,334 |
| **PPI** | 56,944 | 818,716 | 50 | - | 121 | 44,906/6,514/5,524 |
| **Two Circles** | 4,000 | 10,000 | 2 | - | 2 | 80/80/3,840 |
| **BA-High** | 200 | 2000 | 50 | (0,1) | 2 | 10/10/180 |

The Two Circle dataset is visually presented in Figure 7 (test accuracies are shown in the corner). The coordinates of each data point are used as features.

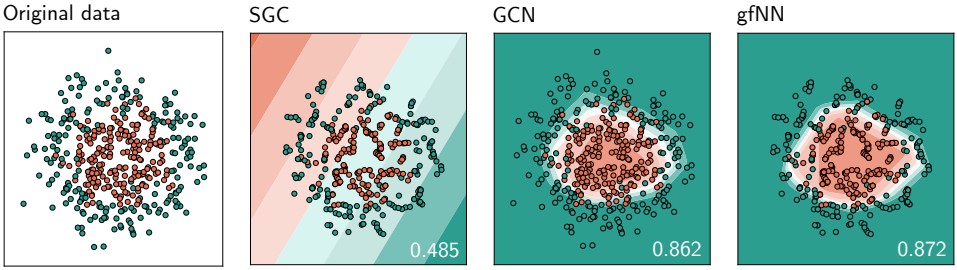

Figure 7: Decision boundaries on 500 generated data samples following the two circles pattern

BA-High dataset generation procedure is described in Figure 8.

We have similar experimental results on different type of random graphs: ER, Configuation, and Block Model. The experimental results on GCN/SGC can be explained simply with Figure 9

In Figure 9, the color blue and orange indicate label 1 and label 0 in a 100 node BA-graph. It is trivial to see that any simple machine learning model can learn from the original feature. However, after only one filter, the features become indistinguisable. More low-pass filter applications will lead to a uniform feature (Theorem 3).

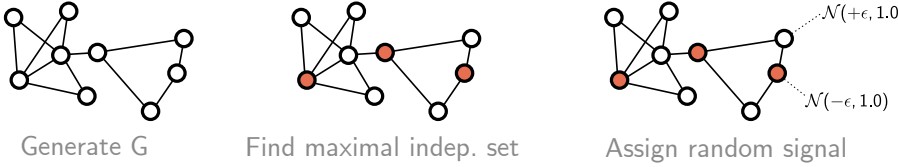

Figure 8: BA-High generation steps

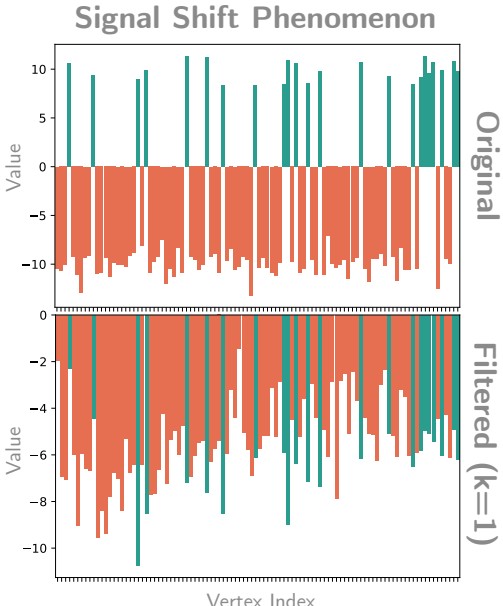

Figure 9: Plot for 1 dimensional features in BA-High setting. We set $\epsilon = 5.0$ for clarity purpose.

