# OpenReview forum: "Frequency Analysis for Graph Convolution Network"
_ICLR.cc/2020/Conference — Reject_

### Official Review · AnonReviewer2 · 2019-10-22
**Official Blind Review #2**

**Rating:** 6

**Review:**

The paper shows the graph signal convolution in GCN-based models is typically a low-pass filter. Besides, the authors propose a simplified GCN-framework named gfNN, which is demonstrated on various benchmark datasets.  To be honest, the paper is well-motivated, well-written. And,  I enjoyed reading through the paper.

However, I have some concerns regarding the technical contributions of this paper.
(1) "GNN is a low-pass filter" is not a new observation. Various works (SGC, [Shuman, et al., 2013]) have studied this observation before.
(2) If the frequency of the label vector is higher than the threshold of the low-pass filter (e.g., gfNN), will the model be problematic?
(3) It seems the only difference between SGC and gfNN is that gfNN is equipped with one more hidden layer. The authors claim that gfNN perform better than SGC is because "the true features are non-linearly separable". If it is the case, then the technical contribution of gfNN is trivial.

Overall, I still like this paper due to its interesting point of the presentation. I vote for weak accept, but I am fine if it is rejected.

**Experience Assessment:**

I have published one or two papers in this area.

**Review Assessment: Checking Correctness Of Derivations And Theory:**

I carefully checked the derivations and theory.

**Review Assessment: Checking Correctness Of Experiments:**

I carefully checked the experiments.

**Review Assessment: Thoroughness In Paper Reading:**

I read the paper thoroughly.

---

> ### Author Response · Authors · 2019-11-06
> **The data itself has low frequency.**
>
> Hi,
>
> Thank you very much for spending your time with our paper.
>
> (1) As in our paper, we addressed the previous observation and stated that our proof for "low-pass filtering" has the same results as in SGC, but using a different technique. In addition, our paper suggests why low-pass filtering works by looking at the Rayleigh quotient of the data itself, which would be helpful to analyze a dataset before using GCN/SGC (graph filtering techniques).
>
> (2) Yes, if the frequency of the label vectors are higher, low-pass technique wouldn't work. We show that in Figure 6. However, to be honest, we couldn't find a real-world dataset that exhibit this high frequency nature so we have to generate a synthetic experiment. As you can see in our Fig 6 (also fig 9), even if the features are very "informative", applying low-pass filtering technique will make the problem harder. Surprisingly, we found that even more complex methods do not work in this "high-frequency" case.
>
> (3) We agree that our contribution of gfNN is trivial, but we wrote this paper because we think the implication in analysis would be interesting and helpful and gfNN is a tool to do that empirically. Fig 7 shows how problem might arise when we use SGC. We wanted to say that the low-frequency nature of data influence the design of Graph Neural Nets, hence we need to look more carefully in to the graph structure (might be represented by filter frequency) and the convexity of the node features themselves.
>
> Thank you again for your time!

---

### Official Review · AnonReviewer3 · 2019-10-23
**Official Blind Review #3**

**Rating:** 1

**Review:**

The authors extend the existing work SGC [1] to a nonlinear version, which addresses the limitations of dealing with nonlinear feature. It further extends the theoretical finding in [1] about the low-pass filtering functionality of graph convolutional networks and shows its advantage in dealing with graph signal processing problem. Evaluation of the proposed method is performed on 7 datasets for node classification task.
Pros:
1. This work goes into detail of the theoretical finding of SGC.
2. Authors conduct extensive experiments on multiple datasets.
Cons:
1. The proposed graph neural network mode is an extension of the existing model SGC, which address the limitation of SGC to model the node feature nonlinearity. It is a good extension, but the novelty is limited.
2. Authors further extend the theoretical finding in [1] and verify the fact that low-pass filtering functionality of graph neural network provide better noise robustness than two layer MLP. It has some novelty, but the novelty is incremental.
3. In the experiment, the performance gap between SGC and gfNN is very small on noise robustness study and traditional node classification. The performance gap between GCN and gfNN is also small on noise robustness study is also small, which can not support the conclusion "GCN has a risk of overfitting to the noise"
4. The paper has the problem of notation missing, e.g on page 2, Fig1 is never mentioned; on page 7, the notation of "LG" is missed in Fig 5.
5. The code link is given but no code is missing.
[1]Wu, Tianyi Zhang, Amauri Holanda de Souza Jr., Christopher Fifty, Tao Yu, and Kilian Q.Weinberger. Simplifying graph convolutional networks. ICML2019


**Experience Assessment:**

I have published one or two papers in this area.

**Review Assessment: Checking Correctness Of Derivations And Theory:**

I carefully checked the derivations and theory.

**Review Assessment: Checking Correctness Of Experiments:**

I carefully checked the experiments.

**Review Assessment: Thoroughness In Paper Reading:**

I read the paper thoroughly.

---

> ### Author Response · Authors · 2019-11-06
> **We apologize about the code link and we made some update following your comments**
>
> Hi,
>
> Update: We updated our manuscript according to your comment (4) and (5). Thank you very much!
>
> Thank you for reading our paper. Please let us address the "cons" part of your comment.
>
> (1) You are right. The novelty of gfNN itself is trivial. However we find gfNN is a very useful analysis tool. Originally (not in the paper), this model comes from our analysis of how important features are to GCN. When we asked the question: what if the features are not linearly separable, we found that GCN can still work, but SGC failed. Wanting to preserve the simplicity of SGC while dealing with complex feature, we come up with gfNN. Further investigation of SGC and GCN nature lead to this manuscript. We also want to point out our paper did the exact analysis on frequency component in Fig 3, 4, and 6, which is useful for addressing the dataset and design adaptive filters.
>
> (2) It is indeed a very small incremental step in research.
>
> (3) On Cora, the performance gap is up to ~10 percent (noisy setting). It is true that same phenomenon is not so clear on other dataset like Citeseer and Pubmed. However, SGC-like method shows better noise resistance to GCN, which we think interesting enough for further study on our claim.
>
> (4) and (5) We apologize for such negligent. We will make the clean version of the code available after this reviewing process. The Colab link to run experiment was harder to make anonymous that we initially thought. We should have provided a zipfile link (included datasets): https://gofile.io/?c=JrE62o. After install dependencies (networkx, pytorch==1.0), run 'python citation.py --model gfnn --dataset cora'.
>
> Thanks again for giving comments to our paper.

---

### Official Review · AnonReviewer1 · 2019-10-23
**Official Blind Review #1**

**Rating:** 6

**Review:**

Summary

Several theoretical analyses have been conducted on MPNN-type NNs. On the other hand, although many graph neural network models take the approach of graph filtering, there is little theoretical analysis on graph NNs that take the filtering approach. This paper analyzed the learnability of filtering type graph NNs.
First, the authors hypothesized that the data used practical tasks consisted of true low-frequency information and high-frequency noise (Assumption 2), and verified the correctness of the assumption.
Next, the authors quantitatively evaluated the difference between the case where we feed noisy data to a graph NN and the case where we feed noiseless data to an MLP. The authors showed that a two-layer GCN approximately behaves as a noise-filtering + a two-layered MLP (Theorem 8). Based on this observation, the authors proposed gfNN, which directly models the latter architecture. We can interpret gfNN as an SGC whose final linear transformation is replaced with an MLP.
Finally, the authors empirically showed that gfNN achieved the same level of accuracy as existing GNNs in the citation network tasks. Furthermore, the authors created a dataset that has meaningful information in the high-frequency area (BA-High). It was shown that the existing GNN, including gfNN, did not perform well, while a GNN designed to pass high frequency (gfNN-high) can predict accurately.


Decision

I judge that this paper contributes to deepning the understanding of graph NNs and is worth to be accepted based on the following three points.
First, it enabled us to understand what causes the oversmoothing phenomena. Several studies have shown that Laplacian-type graph convolution works as a low pass filter. However, most of them considered a linear setting and did not explain how the graph convolution behaves when a graph NN has non-linear activation functions. Compared to them, their result admits non-linearity.
Second, by showing that we can approximate a GCN by a noise filter followed by an MLP, this paper has made the relationship between a GCN and an SGC clearer.
Finally, the authors experimentally showed that existing graph NNs do not have predictive power when useful information is in high-frequency domains. It gives insights on what graph NNs can and cannot solve.
For these reasons, I think this paper has contributed to a deeper understanding of graph NNs and sufficiently significant.


Suggestions

- If I understand correctly, the title of Section 4 reflects the content of Lemma 5 in Section 5 rather than the content of Theorem 3 and (5). I recommend the authors to reconsider the titles of Sections 4 and 5.
- Could you add more explanations to the proof of Theorem 8 in Appendix A. Especially, I could not understand how the term $\tilde{O}(\epsilon^{1/4})$ is derived.


Questions

- I could not find implementations of graph NNs in the notebook to the code (I only found the ls result of dataset directories). Do you plan to release the experiment code?
- At the beginning of Section 6, the authors wrote that they set $k$ to $2$ (hence two-layered GNNs are in consideration). But Theorems 7 and 8 considered the optimal $k^\ast$ in Corollary 6. Which is correct?
- In Section 3, the authors claimed that the performance of MLPs is relatively more robust to the Gaussian noise in the low-frequency regime compared to the high-frequency regime. Certainly, the decrease in performance at $\sigma = 0.01$ is massive in the high-frequency setting for the CiteSeer dataset. However, it is hard to see this trend in Cora and Pubmed. Therefore, I think it is a little too aggressive to conclude that claim.

**Experience Assessment:**

I have published one or two papers in this area.

**Review Assessment: Checking Correctness Of Derivations And Theory:**

I assessed the sensibility of the derivations and theory.

**Review Assessment: Checking Correctness Of Experiments:**

I assessed the sensibility of the experiments.

**Review Assessment: Thoroughness In Paper Reading:**

I made a quick assessment of this paper.

---

> ### Author Response · Authors · 2019-11-12
> **We made some minor updates to our manuscript following your comments**
>
> Hi,
>
> Thank you very much for your comments and questions!
>
> > Firstly, we do plan to have the code available, you can find the core models (including SGC, GCN and noisy settings plus synthetic data) which we use for all experiments here:  https://gofile.io/?c=JrE62o
>
> > Suggestion regarding Theorem 8 (we updated the appendix explaining it in a bit detailed).
> To recap here, the reason for the term $\epsilon^{1/4}$ is from the square root  of Lemma 10 which bounds the D-norm of difference between an activated signal $\sigma(X)$ and a band-limited signal $Y$. This lemma is meant to prepare for Theorem 8, where $\sigma(X)$ is the activation of the *true* noiseless sigmal and $Y$ is the filtered version of the observed signal.
>
> > Question about the value k in Section 6.
> The $k$ under analysis here is 2 as we stated from the beginning. We have to assume $k^*$ is 2 and furthermore we assume this value comes from Corollary 6. Since GCN couples the number of MLP layers and k, we find it unfair to analyze some other values for k in case of SGC and gfNN. We plan to improve this argument in later version of our manuscript.
>
> > Question about frequency component experiments in Section 3 (Figure 3).
> Can we ask are you referring to Figure 5 (Section 7) instead of Figure 3 (Section 3)? We think our Figure 3 clearly shows our Assumption 2 (low-frequency assumption) experimentally across all data set. Even at 0.01, when we use all frequency components, all datasets show more than 10% decrease in performance. Please let us know if you meant something else.
>
> > Suggestion about the title of Sections 4 and 5.
> Thank you very much. We will consider to change accordingly to make it clearer to reader.
>
> Thank you again for your time and your comments!

---

> > ### Comment · AnonReviewer1 · 2019-11-13
> > **Explanation about my comment on Figure 3**
> >
> > >> Can we ask are you referring to Figure 5 (Section 7) instead of Figure 3 (Section 3)?
> >
> > I meant Figure 3. I did not agree with the following sentence:
> >
> > > By adding artificial Gaussian noise, we observe that the performance at low-frequency regions is relatively robust, which implies a strong denoising effect.
> >
> > Take the case where we use approximately 1000 frequency components in the PubMed dataset. I thought that the difference in performance between the $\sigma=0$ case and the $\sigma=0.05$ case is large. We can say the same thing in the case of approximately 400 frequency components in Cora and 500 frequency components in CiteSeer.
> > I understand that the performance drops in these situations are certainly relatively smaller than the situations in which we use all frequency components. However, I was not sure we could safely say that this observation has justified Assumption 2.
> >
> > I would appreciate if authors let me know if my explanation is still unclear.

---

> > > ### Author Response · Authors · 2019-11-14
> > > **About Figure 3**
> > >
> > > Hi,
> > >
> > > Thank you for clarifying your comment to us! It is clear now.
> > >
> > > We think you have a point here. It is indeed a large dip even within the low frequency domain for all datasets at $\sigma = 0.05$. From your comment, we agree that our claim for "strong denoising" might be too aggressive. We can say there is a denoising effect and provide some more perspective.
> > >
> > > To put in perspective, here are the standard deviation and mean of the signal $X$ for each dataset (across all channels):
> > > - Cora: $\mu_X = 0.0007$; $\sigma_X = 0.0071$
> > > - Citeseer: $\mu_X = 0.0003$; $\sigma_X = 0.0029$
> > > - Pubmed: $\mu_X = 0.002$; $\sigma_X = 0.0087$
> > > If we consider only the non-zero entries:
> > > - Cora: $\mu_X = 0.0311$; $\sigma_X = 0.05$
> > > - Citeseer: $\mu_X = 0.03$; $\sigma_X = 0.0065$
> > > - Pubmed: $\mu_X = 0.02$; $\sigma_X = 0.02$
> > >
> > > As you can see, noise level $\sigma = 0.01$ and $\sigma = 0.05$ is extremely large here. The reason for us to choose such large noise level is to support for the "strong denoising" argument. Nonetheless, we will make the argument clearer and more exact.
> > >
> > > Thank you very much for your time and your valuable comment!

---

> > > > ### Comment · AnonReviewer1 · 2019-11-14
> > > > **Thank you for clarification**
> > > >
> > > > I thank the authors for considering my comment seriously and for providing additional information for clarification.
> > > >
> > > > I think the statistics of the datasets is informative for interpreting the noise level $\sigma=0.01, 0.05$. Although it is up to the authors, I recommend to add the discussion in the previous comment to the paper.

---

> > > > > ### Author Response · Authors · 2019-11-14
> > > > > **Thank you!**
> > > > >
> > > > > Update: We add a footnote to the paragraph and extra information of the three datasets in Table 2 (appendix).
> > > > >
> > > > > Indeed, we will update our manuscript with this point. Thank you again for pointing this out!

---

### Decision · Program_Chairs · 2019-12-19

**Decision:**

Reject

**Comment:**

This paper studies two-layer graph convolutional networks and two-layer multi-layer perceptions and develops quantitative results of their effect in signal processing settings. The paper received 3 reviews by experts working in this area. R1 recommends Weak Accept, indicating that the paper provides some useful insight (e.g. into when graph neural networks are or are not appropriate for particular problems) and poses some specific technical questions. In follow up discussions after the author response, R1 and authors agree that there are some over claims in the paper but that these could be addressed with some toning down of claims and additional discussion. R2 recommends Weak Accept but raises several concerns about the technical contribution of the paper, indicating that some of the conclusions were already known or are unsurprising. R2 concludes "I vote for weak accept, but I am fine if it is rejected." R3 recommends Reject, also questioning the significance of the technical contribution and whether some of the conclusions are well-supported by experiments, as well as some minor concerns about clarity of writing. In their thoughtful responses, authors acknowledge these concerns.  Given the split decision, the AC also read the paper. While it is clear it has significant merit, the concerns about significance of the contribution and support for conclusions (as acknowledged by authors) are important, and the AC feels a revision of the paper and another round of peer review is really needed to flesh these issues out.